# Virulence, Antibiotic Resistance and Cytotoxic Effects of *Lactococcus lactis* Isolated from Chinese Cows with Clinical Mastitis on MAC-T Cells

**DOI:** 10.3390/microorganisms13071674

**Published:** 2025-07-16

**Authors:** Tiancheng Wang, Fan Wu, Tao Du, Xiaodan Jiang, Shuhong Liu, Yiru Cheng, Jianmin Hu

**Affiliations:** College of Animal Science and Veterinary Medicine, Shenyang Agricultural University, Shenyang 110866, China; tianchengwang@syau.edu.cn (T.W.); m15904926531@163.com (F.W.);

**Keywords:** bovine clinical mastitis, *Lactococcus lactis*, antimicrobial resistance, pathogenicity, virulence gene

## Abstract

*Lactococcus lactis* (*L. lactis*) is a pathogenic Gram-positive, catalase-negative coccobacillus (GPCN) associated with bovine mastitis. In this study, nine strains of *L. lactis* were successfully isolated and characterized from 457 milk samples from cows with clinical mastitis in China. All isolates exhibited a high degree of susceptibility to marbofloxacin and vancomycin. A series of molecular and cell biological techniques were used to explore the biological characteristics and pathogenicity of these isolates. The virulence gene profiles of the isolates were analyzed using whole genome resequencing combined with polymerase chain reaction (PCR) to elucidate the differences in virulence gene expression between isolates. To provide a more visual demonstration of the pathogenic effect of *L. lactis* on bovine mammary epithelial cells, an in vitro infection model was established using MAC-T cells. The results showed that *L. lactis* rapidly adhered to the surface of bovine mammary epithelial cells and significantly induced the release of lactate dehydrogenase, suggesting that the cell membranes might be damaged. Ultrastructural observations showed that *L. lactis* not only adhered to MAC-T cells, but also invaded the cells through a perforation mechanism, leading to a cascade of organelle damage, including mitochondrial swelling and ribosome detachment from the endoplasmic reticulum. The objective of this study was to provide strong evidence for the cytotoxic effects of *L. lactis* on bovine mammary epithelial cells. Based on this research, a prevention and treatment strategy for *L. lactis* as well as major pathogenic mastitis bacteria should be established, and there is a need for continuous monitoring.

## 1. Introduction

Mastitis in dairy cattle represents a significant obstacle to the advancement of the dairy farming industry, with a high prevalence across the globe [1]. Bovine mastitis is believed to be caused by contagious or environmental bacteria including *Staphylococcus aureus*, *Streptococcus agalactiae*, *Escherichia coli*, *Klebsiella pneumonia*, *Enterobacter aerogenes,* and *Mycoplasma bovis* [2]. The etiology of this disease is multifactorial, with pathogenic microorganism infection serving as a pivotal element. Notably, there has been a long-standing association between *Lactococcus* spp. and mastitis, with the first such association being reported in 1932 [3]. However, the phenotypic and biochemical similarities between Gram-positive catalase-negative (GPCN) *Streptococci* and *Streptococci*-like bacteria, including *Streptococcus*, *Enterococcus*, *Aerococcus*, and *Lactococcus*, present a challenge in accurate identification of *Lactococcus* species. This may result in an underestimation of disease incidence or erroneous diagnoses associated with *Streptococcus uberis* (*S. uberis*) or *Streptococcus* spp. [4]. Despite the under-recognition of the clinical significance of *Lactococcus* spp. as mastitis pathogens, scholars have increasingly focused on these organisms in recent years.

Due to the similarities in their phenotypic characteristics, the genus *Lactococcus* was initially mistakenly included in the genus *Streptococcus* until it was defined as a separate genus in 1985 [5]. *Lactococcus lactis* comprises a diverse population with several subspecies, of which *Lactococcus lactis* subsp. *lactis* and *Lactococcus lactis* subsp. *cremoris* are the most widely used in industrial applications as fermentation agents for various dairy products, including fermented milk and cheese [6]. Although *Lactococcus lactis* is generally considered a food-safe microorganism, it has been detected in diseased poultry, mastitis-affected dairy cows, diseased fish [7], and even in human clinical samples [8]. In particular, the strain has been successfully isolated from the mammary skin of dairy cows, sand bedding materials in feeding environments, and bulk raw milk [9], suggesting that it may be present and transmitted in diverse ecological settings.

Putative virulence genes, such as hemolysis-, adhesion-, and immune-evasion-related genes, and other putative virulence genes of bovine-derived *L. garvieae* have been described [10]. Given the significant genetic homology between *L. garvieae* and *Lactococcus lactis*, this species is an important reference for the study of bovine-derived *Lactococcus lactis* [11]. *Lactococcus lactis* possesses a diverse array of virulence factors, including key genes and proteins such as superoxide dismutase (SOD) [12], hemolysin (hly1, hly2, and hly3) [13], and exopolysaccharide (EPS)-associated genes (*epsR, X*, *A*, *B*, *C*, *D*, and *L*) [14,15]. Of these, EPS has been linked to pathogenicity in mammalian hosts and contributes to the formation of bacterial surface capsules [16]. In addition, phosphoglucomutase (pgm) and NADH oxidase play important roles in bacterial virulence. It is postulated that the virulence of strains lacking the pod gene cluster may be related to virulence factors encoded by SOD, NADH oxidase, phosphoglucomutase, and enolase (eno) [17]. At present, the mechanism of action of *Lactococcus lactis* has not been adequately investigated [18], focusing mainly on phenotypic or genotypic characterization, and there are significant phenotypic differences between pathogenic isolates; therefore, it is crucial to investigate the putative virulence genes of *Lactococcus lactis* strains in depth.

Antibiotics are particularly favored for the managing of mastitis [19]. Many antimicrobial agents, i.e., beta-lactams, tetracyclines, macrolides, aminoglycosides, and fluoroquinolones, are commonly used to treat clinical bovine mastitis in China [20]. Fluoroquinolones and cephalosporins are widely used, particularly third and fourth generation drugs [21]. Fluoroquinolones are extremely effective and used to deal with both Gram-negative and positive bacterial infections in clinics [22]. However, the prolonged or inappropriate administration of these medications can result in the development of bacterial resistance [23], which poses a significant threat to the effective treatment of mastitis. Antibiotics promote the activation of several mechanisms, including efflux pump expression, cell wall recycling, porin reduction, target protein alteration, and biofilm formation, which participate in bacterial resistance to several compounds [24]. The high levels of resistance to various antimicrobial agents have become a real problem in China [25,26]. Consequently, it is essential to monitor resistance patterns among pathogenic bacteria, as this resistance can reduce available treatment options [27,28], increase healthcare expenditures, and heighten complications and mortality rates. Fortunately, research on alternative therapeutic approaches to antibiotics has made progress, such as the use of antimicrobial peptides and macrocycles, bacteriophages, and natural antimicrobial compounds. For instance, essential oil from *Thymus vulgaris* (TEO), a natural antimicrobial compound, was reported to be effective in combating mastitis-associated bacteria [29]. The objectives of this study were to elucidate the epidemiology, antimicrobial resistance profiles, and potential virulence genes of *L. lactis* isolates from preliminary mastitis samples.

## 2. Materials and Methods

### 2.1. Farm Distribution and Clinical Mastitis Sample Collection

CM milk samples (*n* = 457) were obtained from 5 Chinese commercial dairy farms from 5 provinces [30]. Samples were systematically collected on a 7-day cyclical schedule for each farm, employing aseptic techniques to ensure quality and integrity. Subsequently, the samples were rapidly frozen at −20 °C and transported via a cold chain to Shenyang Agricultural University in China for comprehensive strain identification and analysis.

### 2.2. Bacterial Isolation and Culture

The collected clinical milk samples were subjected to separation and purification using 5% sheep blood agar and brain heart infusion (BHI) agar. Specifically, 10 μL was inoculated onto 5% defibrinated sheep blood agar plates and incubated at 37 °C for 24 h. Following the incubation period, the isolated bacteria were subsequently cultured in BHI broth. The morphology of *L. lactis* was then evaluated through gross observation, Gram staining, and scanning electron microscopy (SEM).

### 2.3. 16S rDNA Sequencing for Identification and Biochemical Detection

Single colonies of *L. lactis*, initially identified morphologically, were isolated and inoculated into BHI medium, followed by incubation on a shaker at a controlled temperature of 37 °C for 18 to 24 h. Subsequently, bacterial DNA was extracted using a commercially available DNA extraction kit from Sangon Biotech (Shanghai) Co., Ltd., Shanghai, China. The extracted DNA was employed as a template for polymerase chain reaction (PCR) amplification, with the identification of the isolates confirmed through 16S rDNA sequencing. PCR amplification employed primers p27f (5′-AGAGTTTGATCCTGGCTCAG-3′) and 1492r (5′-TACGGCTACCTTGTTACGACTT-3′) to generate a 1460 bp fragment of the 16S rDNA gene [31]. The resulting PCR products subsequently underwent Sanger sequencing, which was conducted by Sangon Biotechnology (Shanghai) Co., Ltd., Shanghai, China. The 16S rDNA sequences obtained were submitted to the nucleotide database of the National Center for Biotechnology Information (NCBI) to conduct a BLAST (v2.12) analysis in order to determine the degree of similarity between the sequences and those already stored within the database. To further assess the physiological and biochemical characteristics of the strains, eleven biochemical tests were conducted on nine *L. lactis* isolates. The biochemical reaction kits used in these tests were supplied by Qingdao Hi-Tech Industrial Park HaiBo Biotech Co., Ltd., (Qingdao City, China). These tests evaluated the fermentation potential of the isolates with respect to ribose, sucrose, lactose, liquid gelatin, sorbitol, maltose, heptulose, galactose, and glucose, in addition to examining the Voges–Proskauer (VP) reaction and alginate utilization. After isolating single colonies on BHI agar (BHIA), the isolates were transferred to ampoules and incubated for the requisite duration at 37 °C. Adherence to specific procedural guidelines for certain biochemical assays is essential to ensure the accuracy and reliability of the measurements obtained.

### 2.4. Growth Curve of Lactococcus lactis

The growth kinetics of *L. lactis* (LL1), *Staphylococcus aureus* (*S. aureus*), and *Enterococcus faecalis* (*E. faecalis*) were subjected to systematic analysis. The standard strains of *S. aureus* and *E. faecalis* were sourced from the Clinical Laboratory of the College of Animal Science and Medicine at Shenyang Agricultural University. A 30 µL volume of LL1, cultured overnight, was inoculated into 3 mL of sterile BHI medium and incubated with shaking at 37 °C for 18 to 24 h. *S. aureus* and *E. faecalis* were employed as control organisms and maintained under identical treatment conditions. The optical density (OD) of each bacterial suspension was determined using a UV spectrophotometer (Shimadzu Corporation, Kyoto, Japan) at 600 nm [32]. The measurements were taken at 0.5 h intervals during the initial 4 h after inoculation, followed by 2 h intervals for the subsequent 20 h. The protocol yielded 19 evaluations over a 24 h period. For each bacterial suspension, three replicates were established at each specific time point. Each experimental procedure was conducted in triplicate, and the collected data were subsequently employed to construct standard growth curves.

### 2.5. Determination of Antimicrobial Resistance

The minimum inhibitory concentration (MIC) values were determined by the broth microdilution method using cation-adjusted Mueller Hinton broth (MHB) as the medium for all antimicrobial agents [33], with concentrations ranging from 0.03 to 16 μg/mL. Nine strains of *L. lactis* were tested for resistance to ten antimicrobial agents: penicillin (β-lactams), cephalexin (β-lactams), ampicillin (β-lactams), ceftiofur (β-lactams), cefquinome (β-lactams), lincomycin (lincosamide class), oxytetracycline (Tetracycline class), marbofloxacin (Quinolone class), rifaximin (Rifamycin class), and vancomycin (Glycopeptides class). To determine MIC values, *L. lactis* strains were incubated in MHB overnight with shaking. Antibiotics were diluted in MHB and each concentration was tested in triplicate for each strain. The cultures were then incubated at 37 °C for 20 to 24 h. At the conclusion of the incubation period, the cultures were assessed for growth inhibition, with the highest antibiotic concentration that effectively inhibited bacterial growth recorded as the MIC. The quality control strain was the standard strain of *Staphylococcus aureus* ATCC 29213. Quality control parameters were established according to Clinical and Laboratory Standards Institute (CLSI) guideline M100-M07 (2019), and all experimental procedures were performed in triplicate.

### 2.6. Primary Screening for Virulence Factors

Multiplex polymerase chain reaction (PCR) analysis of DNA purified from nine *L. lactis* isolates was performed to assess the expression of 30 key virulence factors with the primers shown in Table 1, according to Xie et al. [11]. These genes include *hly1*, *hly2*, and *hly3*; NADHO; SOD; phosphoglucomutase (pgm); Pav; PsaA; eno, containing surface proteins-1, -2, -3, and -4 (LP1, LP2, LP3, and LP4) [17]; AC1; AC2; Adhesin (Adh); capsule gene cluster (1020-F, 1323-R); capsule gene cluster (851-F, 1399-R); capsule gene cluster (6329-F, 7175-R); capsule gene cluster (5358-F, 6007-R); conserved hypothetical protein (CHP); exopolysaccharides R, X, A, B, C, D, and L (epsRXABCDL); oligosaccharide repeat unit polymerase (ORUP); rhamnosyltransferase (RTU); rhamnocyte transferase (RCT) [34]; rhamnosyltransferase (RIF); and the 30S rRNA gene. The total reaction mixture was 25 μL, including 12.5 μL of Green Teq Mix, 1 μL of template DNA, 1 μL of each primer, and 9.5 μL of ultrapure distilled water. The first step in the amplification procedure was denaturation at 95 °C for 5 min. This was followed by a series of cycles including denaturation at 95 °C for 15 s, annealing at 52 °C for 30 s, extension at 72 °C for 60 s and extension at 72 °C for 5 min. The PCR products were analyzed via electrophoresis on a 1.2% (*w*/*v*) agarose gel at 120 V for 35 min [35]. PCR products from no less than three randomly selected strains were analyzed for related genes in the *L. lactis* isolates through a homology search against the NCBI database.

### 2.7. Genome Sequencing, Assembly, Annotation and Bioinformatics Analysis

DNA was extracted from LL1 strain samples using the Illumina DNA preparation kit (Illumina Inc., San Diego, CA, USA) [36]. To construct the library, the bacterial chromosomal DNA of strain LL1 was subjected to mechanical and random cleavage into short-read length fragments, according to the instructions provided by the manufacturer. This library was then subjected to large-scale sequencing on the machine, providing raw data in the FASTQ format [37]. The overlapping regions between the different reads were then spliced to obtain a contig, that is, the complete sequence of the bacterial genome. A paired (PE150) configuration was employed for whole genome resequencing using the Illumina sequencing platform, with ab initio sequence assembly conducted using SPAdes version 3.15.1. The MUMmer software (v3.22) was employed to comprehensively display the genes, repeat sequences, annotation information, GC content, and other pertinent data on the genome of the sequenced strains [38]. The BWA software (v0.7.10) was employed to compare the valid reads (clean reads) of the LL1 samples and the reference genome hg 19 (available from NCBI at https://www.ncbi.nlm.nih.gov/assembly/GCF_000344575.1 accessed on 22 November 2021).

### 2.8. Cell Culture

Bovine mammary alveolar cells (MAC-T) were prepared and cultured as previously described. The cells were maintained in Dulbecco’s Modified Eagle’s Medium (DMEM), supplemented with 10% fetal bovine serum (FBS) and 1% penicillin and streptomycin (from Sigma Aldrich, St. Louis, MO, USA). The culture conditions were set to 37 °C in a 5% CO_2_ atmosphere.

### 2.9. Cell Viability Using the CCK-8 Method

Well cultured MAC-T cells were seeded at a density of 1 × 10^4^ cells per well in 96-well plates (Corning Inc., Corning, NY, USA) and incubated under standard conditions. Subsequently, the LL1 strain was introduced to the respective wells at multiplicities of infection (MOI) of 5, 25, 50, 100, and 200. The plates were incubated for time intervals of 1, 3, 6, and 12 h. A control group was maintained, and each condition was replicated in three independent trials to ensure the statistical robustness of the results. Following the manufacturer’s protocol, 10 μL of CCK-8 reagent was added to each well. The plates were gently agitated to ensure thorough mixing and then further incubated for 1 h. The post-incubation optical density of each well was measured at a wavelength of 450 nm to assess cell viability.

### 2.10. Lactate Dehydrogenase (LDH) Release Assay

The cytotoxic effect of *L. lactis* on the bovine mammary alveolar cell line T (MAC-T) was evaluated using a lactate dehydrogenase (LDH) assay kit (Beyotime Biotechnology, Beijing, China). MAC-T cells were cultured in 96-well plates (Corning Inc., Corning, NY, USA) until they reached approximately 80% confluence. The cells were then exposed to *L. lactis* at a multiplicity of infection (MOI) of 5:1 for designated time points of 1, 3, 6, 12, and 24 h. After incubation, LDH activity in the MAC-T cells was quantified according to the manufacturer’s protocol for the assay kit. Absorbance was measured at 490 nm using a microplate reader (QuantStudio3, Thermo Fisher, Waltham, MA, USA). Each experimental condition was performed in triplicate to ensure the reliability and reproducibility of the data.

### 2.11. Adhesion and Invasiveness Measurements

MAC-T cells were cultivated in 96-well plates and pretreated with 4% FBS double antibody-free medium until the cells reached approximately 80% confluence. LL1 was introduced at a multiplicity of infection (MOI) of 50:1 for varying incubation periods of 30 min, 1 h, 2 h, and 3 h. The control group underwent the same experimental conditions, with three replicates for each time point. After the designated incubation period, the cells were subjected to two washes with sterile phosphate-buffered saline (PBS) (pH 7.4). To release adherent bacteria and lyse the cells, 2 mL of 0.5% (*v*/*v*) Triton solution was added. By contrast, the control group received 2 mL of 1% (*v*/*v*) Triton X-100 solution directly. The collected bacterial suspension was then serially diluted tenfold. The diluted samples were inoculated onto BHIA and incubated at 37 °C for 24 h to facilitate bacterial growth. Colony forming units (CFUs) were enumerated and the number of colonies and corresponding CFU values were recorded. Three replicates were performed on each test plate. The adhesion rate of LL1 to MAC-T cells was calculated using an appropriate formula. If LL1 exhibited invasive properties, further studies were warranted to investigate the effect of bacterial invasion on the adhesion rate.

To assess the invasiveness of LL1 on MAC-T cells, the previously described culture process was carefully followed. Each well of the experimental group was supplemented with 1 mL of penicillin medium diluted in 4% FBS, while the control group received an equivalent volume of saline. The cultures were then incubated for 1 h at 37 °C in a 5% CO_2_ atmosphere to extract extracellular bacteria. In the experimental group, 2 mL of 0.5% (*v*/*v*) Triton X-100 solution was added to release intracellular LL1. The same treatment was applied to the background concentration of the bacterial strains. The resulting mixture was allowed to stand for 10 min and then gently vortexed to ensure complete homogenization; the bacterial suspension was then collected, serially diluted and inoculated onto BHIA plates, which were incubated for 24 h to promote colony development. Colony counts and corresponding CFU values were recorded, with each experimental condition replicated three times to ensure statistical reliability.

### 2.12. Cell Morphology and Ultrastructure Analysis

The ultrastructure of MAC-T cells was examined using transmission electron microscopy (TEM). MAC-T cells were cultured at a density of 5 × 10^5^ per well in 6-well plates (Corning Inc., Corning, NY, USA) [39]. Once the culture had reached completion, the cells were incubated with *L. lactis* at an MOI of 5:1 for varying periods, specifically 3, 6, 12, and 24 h. To establish a baseline for comparison, uninfected cells were used as controls, with three replicates established for each time point. After each incubation period, the medium was removed and the MAC-T cells attached to the coverslips were washed three times with sterile PBS. The cells were fixed with 2.5% glutaraldehyde at 4 °C overnight to preserve their morphology. The fixed samples were then pre-embedded in 4% agarose, dehydrated with ethanol, and embedded in a suitable resin for electron microscopy.

## 3. Results

### 3.1. Morphological Characteristics of Lactococcus lactis

*L. lactis* colonies were observed as round, medium-sized colonies on 5% sheep blood TSA plates, measuring approximately 0.5–2 mm in diameter. They exhibited a grayish-white color, smooth edges, moist surfaces, and an uneven texture. Following a 24 h incubation at 37 °C, α-hemolysis was identified, characterized by the presence of a faint green hemolytic ring surrounding the colonies (Figure 1A). Gram staining demonstrated that the bacteria were Gram-positive, exhibiting a purple hue under microscopic examination. Moreover, the organisms displayed a spherical or oval shape, organized in pairs or short chains (Figure 1B). Scanning electron microscopy substantiated the oval morphology of the bacteria, with a diameter of approximately 0.5–2 μm (Figure 1C,D).

### 3.2. Results of 16S rRNA Identification and Homology Analysis

The results of gel electrophoresis for the amplified product are presented in Figure 2A, demonstrating the successful amplification of a band approximately 1500 bp in length. These findings were submitted to NCBI for comparison, which identified the organism as *L. lactis*. A phylogenetic tree was subsequently constructed using *Streptococcus agalactiae* 015-RIA1 and *Escherichia coli* JCM 16,946 as exogenous control strains sourced from the GenBank database, as depicted in Figure 2B. The analysis of the phylogenetic tree revealed that the seven strains of *L. lactis* exhibited relatively close phylogenetic relationships with one another, while they were distantly related to the exogenous control strains.

### 3.3. Biochemical Detection of Lactococcus lactis

The biochemical assays conducted on nine isolates of *L. lactis* demonstrated negative results for all strains in the substrate fermentation assays for ribose, liquid gelatin, and sorbitol. Four isolates (LL2, LL7, LL8, and LL9) exhibited negative outcomes in the sucrose fermentation assay, while the Esculin fermentation assessment revealed that LL2 was negative, with the remaining isolates testing positive. The fermentation assays for lactose, maltose, trehalose, and glucose, along with the Voges–Proskauer test, yielded positive results for all strains. The comprehensive results are provided in Table 2 for reference.

### 3.4. Growth Ability of Lactococcus lactis

Figure 3 illustrates the growth curves of *L. lactis*, *S. aureus*, and *E. faecalis* isolates cultured in BHI broth. The growth profile of *L. lactis* comprises three distinct phases: an initial lag phase of approximately two hours, followed by a logarithmic phase lasting around four hours, and culminating in a stabilization phase. The lag phase exhibited by *L. lactis* mirrored the profile observed in the other bacterial isolates. The growth of *L. lactis* was relatively slow, with a gradual increase in growth rate, peaking at an optical density of approximately 1.0 between 4 and 6 h. After the 6 h mark, no further changes in growth were observed, indicating that the strain had entered a stable phase. By contrast, the maximum optical density values for *S. aureus* and *E. faecalis* were approximately 1.3.

### 3.5. Antimicrobial Resistance Profiles of Lactococcus lactis

As shown in Table 3, all nine *L. lactis* isolates were sensitive to the β-lactam antibiotics marbofloxacin and vancomycin. By contrast, these isolates showed resistance to penicillin, cephalosporin, rifaximin, and hygromycin.

### 3.6. Lactococcus lactis Virulence Factor Primary Screening

The objective of this study was to gain insight into the intricacies of *L. lactis* by examining its capacity to form bacterial pods, its hemolysis-, adhesion-, immune-evasion-related virulence genes, and other putative virulence genes. One hemolysis gene (*hly2*), two adhesion-associated genes (*PsaA* and *AC1*), and a cluster of podopod genes (*CGC C*) were identified. The *Eps* genome demonstrated a total of seven genes coding for exopolysaccharides, including *EpsA*, *EpsB*, *EpsC*, *EpsD*, *EpsL*, *EpsR*, and *EpsX*. These genes play a role in bacterial biofilm formation, immune evasion, and other pathogenic mechanisms. Furthermore, five additional putative virulence genes (*SOD*, *CHP*, *RIF*, *eno* and 30S rRNA) were identified. Figure 4 depicts the hypothetical genetic results found in the CM cases.

### 3.7. Bioinformatics Analysis of the Lactococcus lactis Genome

Gene Ontology (GO) analyses of the LL1 variant genes are illustrated in Figure 5. The variant genes were primarily linked to the overall composition of the membrane, which emerged as the most significantly enriched functional category. This notable enrichment indicates that genetic alterations associated with membrane composition may directly influence the cellular structure and function of *L. lactis*. Additionally, the GO analyses revealed modifications in the functions of mutant genes related to transmembrane transport, methylation, cell wall organization, and various other biological processes.

As illustrated in Figure 6, KEGG (Kyoto Encyclopedia of Genes and Genomes) analysis of the mutated genes in *L. lactis* revealed 51 enriched biological pathways. The metabolic pathway exhibited the highest level of gene enrichment, comprising 328 genes. Additionally, pathways related to secondary metabolites, antibiotics, and amino acid biosynthesis were also notably enriched.

Virulence factors are critical components of bacterial pathogenicity, and this study concentrated on the invasive virulence factors of strain LL1 (Figure 7). Among these factors, toxins accounted for 25.68%, adhesion factors constituted 23.408%, invasion-associated components made up 6.252%, and components of the secretion system represented 44.651%.

### 3.8. Effect of Lactococcus lactis on MAC-T Cell Viability

The CCK-8 assay indicated that infection with LL1 significantly reduced the viability of MAC-T cells compared with the control group (*p* < 0.001). At a multiplicity of infection (MOI) of 5, the viability of the LL1-infected MAC-T cells decreased to approximately 50 within 12 h (Figure 8A). Lactate dehydrogenase (LDH) released from MAC-T cells exposed to LL1 (MOI = 5) at various time points demonstrated significantly elevated LDH levels at 1, 3, 6, 12, and 24 h post-infection when compared with the controls. LDH release was particularly significant (*p* < 0.01) at 6 and 12 h post-infection (Figure 8B).

### 3.9. Pathogenic Effects of Lactococcus lactis on MAC-T

MAC-T cells were subjected to LL1 treatment at various time intervals, revealing a positive correlation between infection duration and both cell adhesion and invasion rates. Specifically, the adhesion rate of the LL1 strain to the surface of the MAC-T cells was relatively low at 1.14% after a 30 min treatment, but this rate significantly increased to a peak of 7.79% following a 180 min treatment period (Figure 9A). In terms of invasion rates, the LL1 strain exhibited an invasion rate of 1.106% at the 30 min mark. Notably, the invasion rate of *L. lactis* on MAC-T cells escalated markedly between 60 and 180 min, culminating in rates of 3.123%, 6.993%, and 10.963% (see Figure 9B).

### 3.10. Morphology of Lactococcus lactis on MAC-T

The ultramorphological alterations in LL1-infected MAC-T cells over time were examined using scanning electron microscopy (SEM) and transmission electron microscopy (TEM) techniques. The results demonstrated that the control cell membranes remained morphologically intact, exhibiting clear internal structures (Figure 10A,B). However, significant morphological degradation was observed in the cells following 3, 6, 12, and 24 h of LL1 treatment. The integrity of the cell surface structure was progressively compromised over time. We observed mitochondrial swelling, marked expansion and fragmentation of the rough endoplasmic reticulum (RER), and weakened ribosome attachment, leading to detachment and depolymerization. Moreover, an increase in bacterial adhesion and the formation of perforated structures were also observed. The nuclei exhibited marginal heterochromatin aggregates and several regions demonstrated extensive disruption of the plasma membrane (Figure 10C–J).

## 4. Discussion

This might be the first time that *L. lactis*, an emerging pathogen associated with mastitis, has been detected in CM samples from China [40]. The presence of *L. lactis* has been verified by molecular techniques, including PCR [41], RAPD-PCR [23], and 16S rRNA sequencing. *Lactococcosis* is relatively common in marine and freshwater farm animals, with prevalence influenced by the environmental conditions of the fish habitat [42]. A variety of environmental variables and feeding management factors can affect morbidity and mortality.

It is also important to note the duality of *Lactococcus lactis*. It is generally considered to be a food-safe microorganism and has been suggested as a relevant bacteria modulator in the food industry, the aquaculture industry, vaccine production, and even for treating mastitis using some strains. In addition, some strains of *L. lactis* were added to the European Food Safety Authority’s Qualified Presumption of Safety (QPS) list, which further clarified its versatility. On the other hand, *Lactococcus lactis* has brought about diseases in various animals as well as in humans [9,43], so it is crucial to clarify the difference between ‘pathogenic’ and harmless strains.

Although *L. lactis* is typically innocuous to healthy individuals, clinical isolates carrying virulence genes may pose a health threat to those with underlying illnesses [44], underscoring the necessity of screening for *L. lactis* as a potential zoonotic pathogen.

Researchers have investigated outbreaks of mastitis in dairy cows and have identified *L. lactis* as a potential causative agent in milk samples from bulk tanks [45]. In recent years, there have been partial encounters with cases of mastitis that have not responded well to treatment [4]. In this study, *L. lactis*-infected farms used sand layers as bedding material. It is hypothesized that sand layers may act as reservoirs for *L. lactis* strains, facilitating the spread of the microorganism among cows via contaminated sand, or be a major source of direct contamination. The relative abundance of *Lactococcus* spp. in the microbiome of mammary gland samples was significantly higher than that observed in milk samples from healthy cows [46].

The findings of the present study’s MIC (Minimum Inhibitory Concentration) test were consistent with those of other studies, which demonstrated that all isolates of *L. lactis* are resistant to penicillin but susceptible to two antibiotics, namely, marbofloxacin and vancomycin [47]. The penicillin resistance exhibited by the nine *L. lactis* strains in this study is at odds with the results for isolates from a related study in southern Germany. Given the prevalent issue of antibiotic abuse in China [48], it was hypothesized that the penicillin use rate in the Chinese pastures where samples were collected might be significantly higher than in the southern German pastures. The variation in penicillin susceptibility among the isolates from these two distinct locations can be attributed to the differential patterns of penicillin use.

All nine *L. lactis* strains possessed the AC1, PavA, PsaA and enolase (eno) genes, which have been implicated in pathogen–host interactions. Despite the significance of hemolysis as a virulence factor, only five strains expressed the hly2 gene, indicating a comparatively low level of virulence. The assessment of critical bacterial surface structures, such as extracellular polysaccharides (EPSs), cellular polysaccharides (CPSs), and lipopolysaccharides (LPSs), is comparable to the EPS biosynthesis gene cluster (eps RXABCD) in all nine strains [49,50]. This finding suggests that podocarp gene clusters are widely disseminated within *L. lactis*. Although the CGC C gene was amplified in three strains, electron microscopy did not reveal any significant cytoarchitectural alterations. Additionally, the expression of superoxide dismutase (SOD) was observed in these strains, which may contribute to maintaining pathogenicity in the absence of the polysaccharide gene cluster [34]. Consequently, a comprehensive evaluation of *L. lactis* pathogenicity necessitates an analysis of multiple virulence genes rather than a narrow focus on a limited few.

Whole-genome resequencing revealed that single nucleotide variants were the most common mutations in *L. lactis* [51]. Comparison with the hg19 genome revealed that mutations in the LL1 strain were primarily located in exonic regions, particularly affecting genes associated with membrane function and metabolic pathways. Significant enrichment was observed in the genes for Asp23 (AmaP) [52], a membrane-anchoring protein, and SecA, a protein involved in adhesion transport; deletion of Asp23 increased cell wall stress, while SecA is involved in the transport of serine-rich glycoproteins [53]. Furthermore, deletion of the 30S ribosomal rpsN gene was closely linked to *L. lactis* metabolism and growth [54]. Metabolically, the MurA, MurB, MurC, and MurD genes, which are crucial for peptidoglycan synthesis, were enriched, and their variants may affect peptidoglycan biosynthesis and assembly, thereby affecting bacterial viability [55]. In addition, the enrichment of eno genes associated with adhesins underlines their role in the pathogenicity of *L. lactis*. Virulence factor analysis revealed a strong enrichment of adhesion-related genes, confirming the results of genetic testing [56]. These findings not only highlight the genetic diversity of *L. lactis* but also confirm that genetic variation related to cell membranes, membrane components, and metabolic pathways is central to the evolution of its pathogenicity.

In this study, we developed a model of *L. lactis* infection in mammary epithelial cells (MAC-T) from dairy cows to investigate the cellular immune response induced by bacterial infection. The CCK-8 assay showed the significant cytotoxic effect of *L. lactis* on MAC-T cells, ultimately leading to cell death. Adhesion and invasion assays showed that *L. lactis* efficiently adhered to and invaded MAC-T cells during the early stages of infection, with an initial adherence rate of 1.14 [57]. This rate increased rapidly over time, peaking within three hours. Scanning and transmission electron microscopy observations revealed significant pathological changes in the nucleus, mitochondria, endoplasmic reticulum, and other organelles of MAC-T cells following infection. At the same time, the bacterial population increased, leading to the disappearance of cellular pseudopods. These findings correlated with elevated LDH levels [58], suggesting that *L. lactis* infection compromises MAC-T cell membranes, resulting in the release of cytoplasmic LDH and highlighting its destructive potential within bovine mammary cells. These results provide new insights and strategies for the prevention and control of mastitis in dairy cows.

Although *Lactococcus lactis* has caused harm as an emerging mastitis pathogen, it can also be treated as a major mastitis-inducing bacterium. For mastitis prevention and control measures, antibiotics are still the first choice but they are limited due to growing bacterial resistance and public health restrictions [19]. However, the development of next-generation antibiotics has been positive, including nanotechnology, antimicrobial peptides and macrocycles, bacteriophages, herbal extracts, probiotics, laser radiation, and natural antimicrobial compounds. In modern and formal farms, professional management of housing, milking, hygienic measures, and teat disinfection are required, and the timely isolation of infected animals to limit the spread of the pathogen is necessary [59]. Finally, for animal health, food safety, and public health, preserving antibiotic efficacy through prudent and targeted use will remain an important focus of future research [60].

## 5. Conclusions

In this study, the zoonotic pathogen *L. lactis* was isolated from cow milk samples sourced from substantial dairy farms in China. All isolates demonstrated sensitivity to marbofloxacin and vancomycin, while exhibiting resistance to penicillin. Whole-genome resequencing revealed the presence of virulence genes associated with transmembrane motility and metabolic processes. Furthermore, a screening of 30 candidate virulence factors identified 16 positive genes present in *L. lactis* from various sources, which were linked to virulence attributes such as adhesion, hemolysis, pod formation, and evasion of host immune responses, indicating potential virulence. *L. lactis* displayed adhesive and invasive capabilities in MAC-T cells, leading to notable disruptions in cellular morphology and structural integrity. The findings elucidate the high prevalence, tissue damage characteristics, and antimicrobial resistance of *L. lactis*, thus laying the foundation for exploring its pathogenic mechanisms.

## Figures and Tables

**Figure 1 microorganisms-13-01674-f001:**
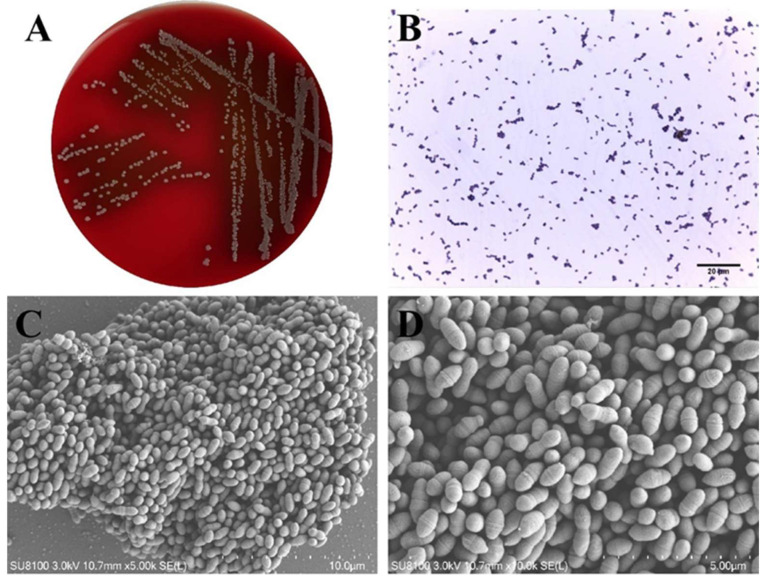
Morphological characteristics of *L. lactis.* (**A**) Pinpoint colonies with α-hemolysis were observed on trypticase soy agar with 5% sheep blood; (**B**) Gram-positive cocci observed under the microscope; and (**C**,**D**) scanning electron microscopy of *L. lactis.*

**Figure 2 microorganisms-13-01674-f002:**
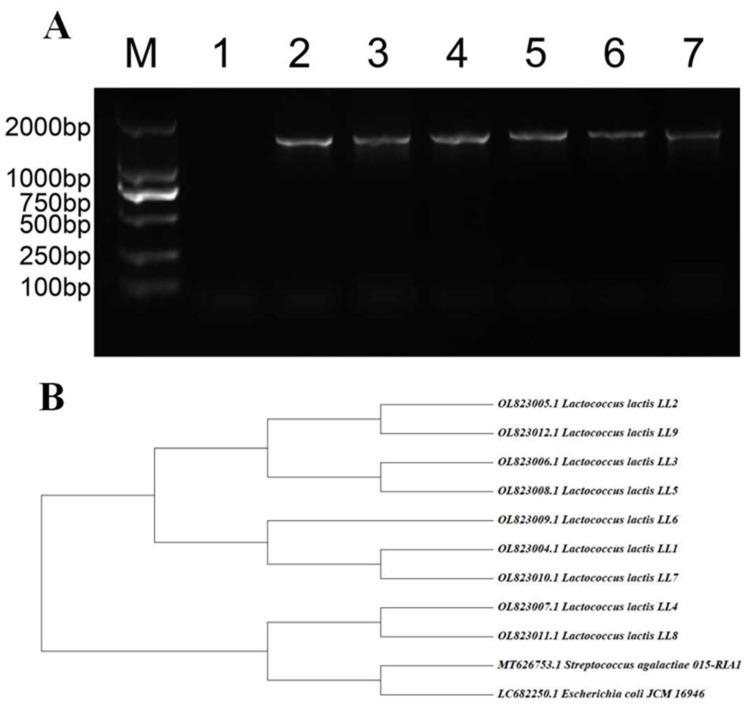
16SrRNA identification and subsequent analysis of homology results. The first column is labeled ‘Mark2000’; the second column is labeled ‘Negative control’; and the third to seventh columns are labeled with the names of the isolated *L. lactis* strains (**A**). These nine *L. lactis* strains were more closely related to each other as a result of comparison with the exogenous control strains (**B**).

**Figure 3 microorganisms-13-01674-f003:**
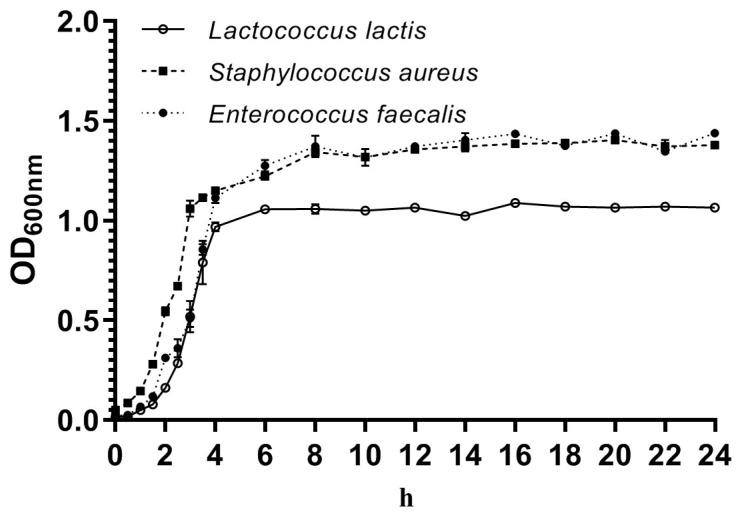
Growth curves of *L. lactis*, *S. aureus*, and *E. faecalis* isolates. Data are mean ± SD of OD600 nm values of three isolates. The curves were plotted using GraphPad Prism 8.

**Figure 4 microorganisms-13-01674-f004:**
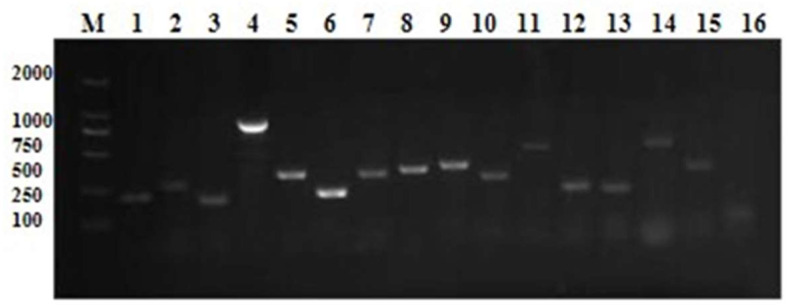
Identification of different virulence genes among *L. lactis* isolates. M: 100 bp DNA ladder. Lanes 1–16: 30S rRNA gene, *AC1*, *PsaA*, *CGC C*, *CHP*, *eno, EpsA*, *EpsB*, *EpsC*, *EpsD*, *EpsL*, *EpsR*, *EpsX*, *hly 2*, *RIF*, and *SOD*.

**Figure 5 microorganisms-13-01674-f005:**
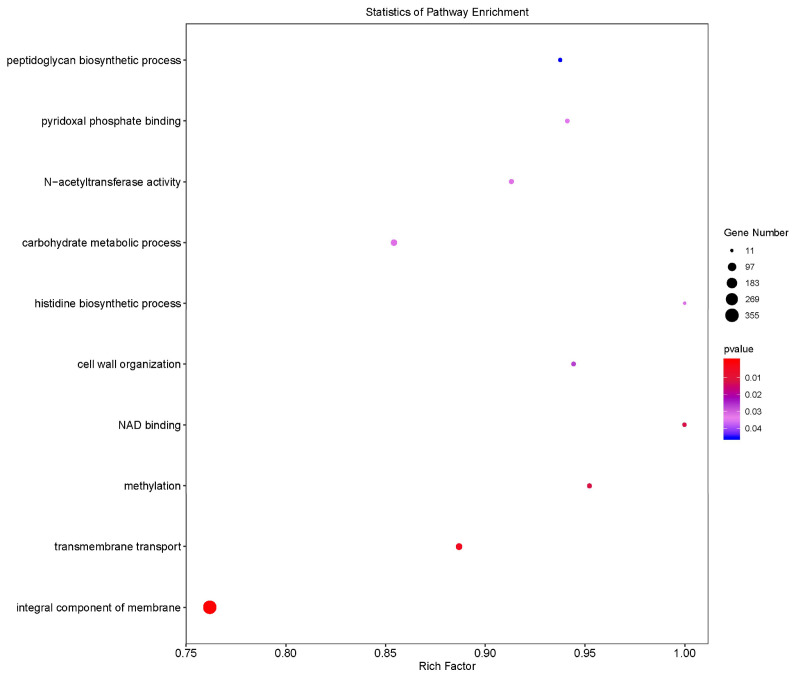
GO analysis of differentially expressed genes in *L. lactis*. LL1 GO results of differentially expressed genes, the size of the dots indicates the number of enriched genes and the color indicates significance.

**Figure 6 microorganisms-13-01674-f006:**
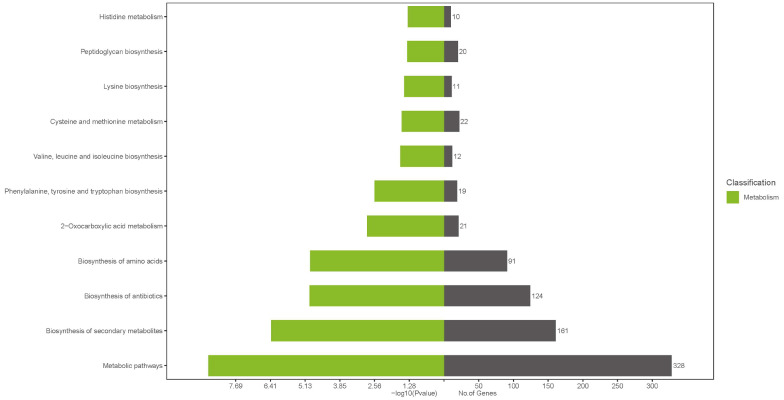
KEGG analysis results of differentially expressed genes in *L. lactis*. The green part on the left side of the figure shows the *p*−value results, reflecting the significance level of each pathway. The black part on the right shows the number of genes involved in each pathway, which intuitively reflects the gene enrichment level of each pathway.

**Figure 7 microorganisms-13-01674-f007:**
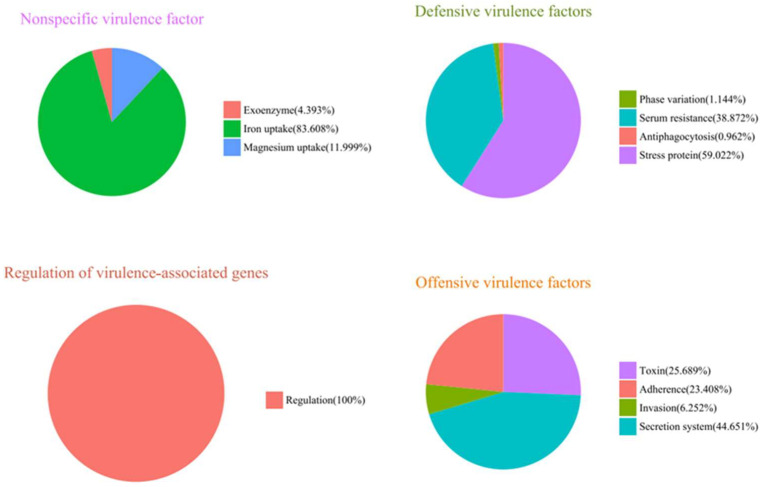
LL1 virulence factor annotated classification statistical results.

**Figure 8 microorganisms-13-01674-f008:**
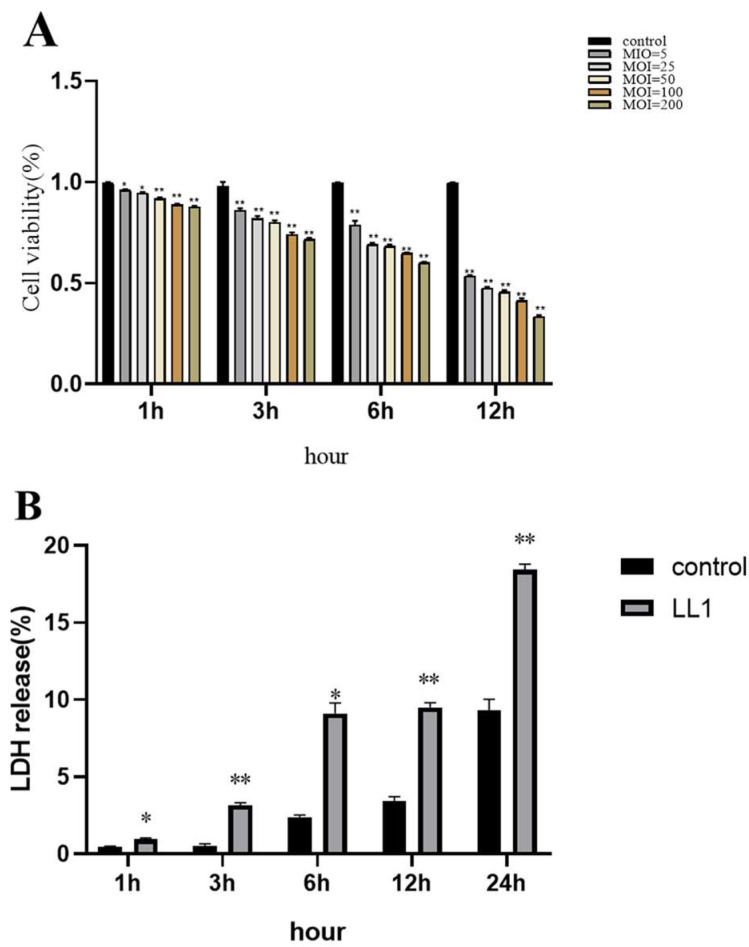
Cytotoxic effects of *L. lactis* infection on MAC-T cells. Effect of *L. lactis* on MAC-T cell viability (**A**). Effect of *L. lactis* on LDH of MAC-T cells (**B**). * indicates a significant different treatment groups (*p* < 0.05 by ANOVA test), and ** indicates a very significant difference between different treatment groups (*p* < 0.01 by ANOVA test).

**Figure 9 microorganisms-13-01674-f009:**
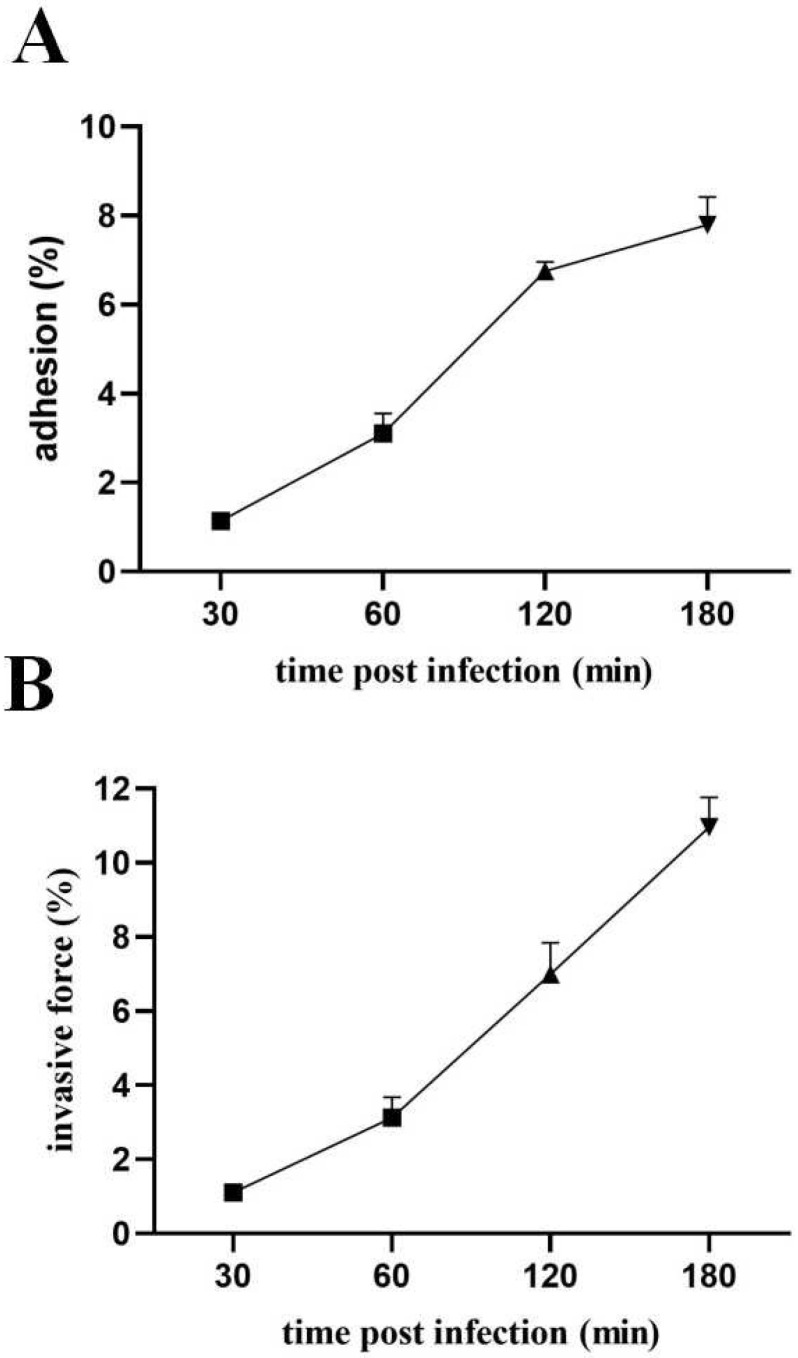
In vitro pathogenic effect of *L. lactis* on bovine mammary epithelial cells (MAC-T). (**A**) Effect of MAC-T on adhesion by *L. lactis* isolates (within 3 h of infection). (**B**) Invasion of MAC-T by *L. lactis*. (within 3 h of infection). Data are mean ± SD of three independent experiments, bar graphs were calculated and plotted using GraphPad Prism 8.

**Figure 10 microorganisms-13-01674-f010:**
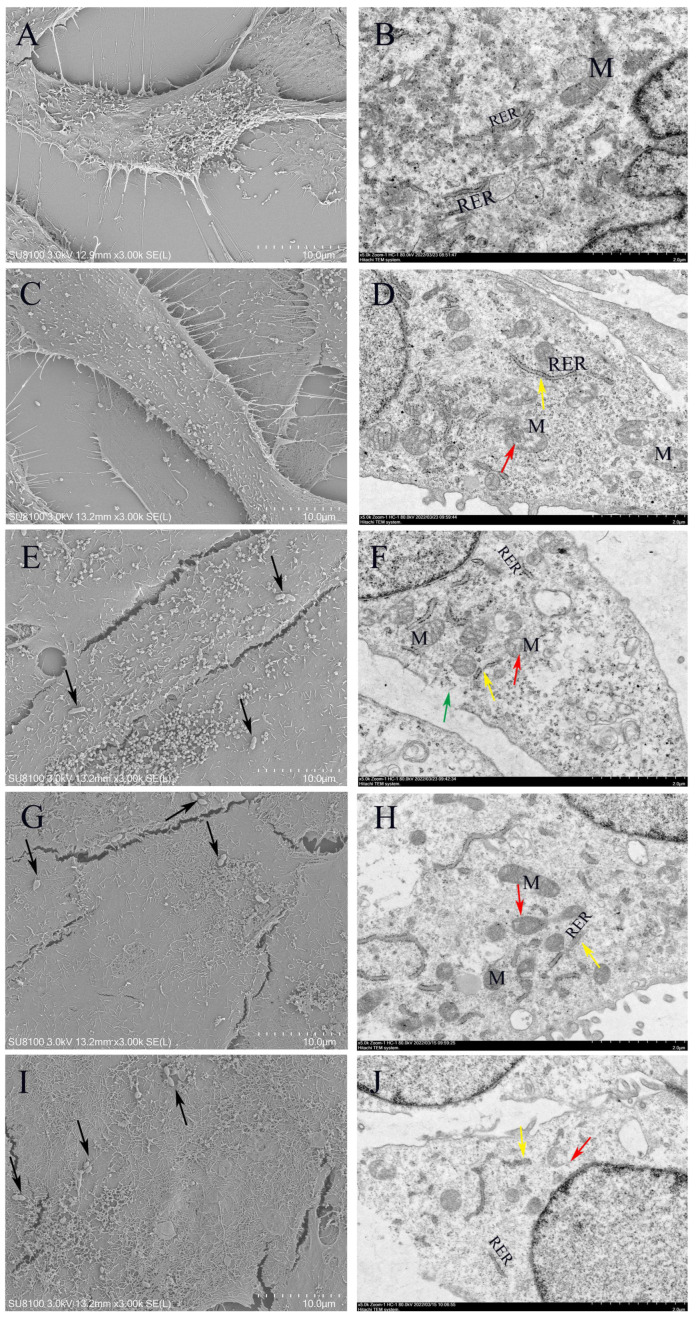
Ultrastructural images of MAC-T infected with *L. lactis* created via scanning and transmission electron microscopy, where black arrows are bacteria, red arrows are mitochondria, yellow arrows are endoplasmic reticulum, and green arrows are cell membranes. LL1 infected with MAC-T (**A**,**B**) control, (**C**,**D**) 3 h after LL1 infection, (**E**,**F**) 6 h after LL1 infection, (**G**,**H**) 12 h after LL1 infection, and (**I**,**J**) 24 h after LL1 infection. M: mitochondria, RER: rough endoplasmic reticulum.

**Table 1 microorganisms-13-01674-t001:** PCR primers.

Target Gene	Abbreviation of Target Gene	Primer Name	Primer Sequence (5′ to 3′)	Product Size	Annealing Temperature (°C)
*Hemolysin 1*	*hly1*	H1 F	CCTCCTCCGACTAGGAACCA	521	54
		H1 R	GAAAAGCCAGCTTCTCGTGC		
*Hemolysin 2*	*hly2*	H2 F	TCTCGTGCACACCGATGAAA	492	53
		H2 R	TGAACTTCGGCTTCTGCGAT		
*Hemolysin 3*	*hly3*	H3 F	AACGCGAGAACAGGCAAAAC	291	56
		H3 R	CCCACGTCGAGAGCATAGAC		
*NADH oxidase*	*NADHO*	NADHO F	TGCGATGGGTTCAAGACCAA	331	53
		NADHO R	GCCTTTAAAAGCCTCGGCAG		
*Superoxide* *dismutase*	*SOD*	SOD F	GCAGCGATTGAAAAACACCCA	80	54
		SOD R	TCTTCTGGCAAACGGTCCAA		
*Phosphoglucomutase*	*pgm*	PG F	AAGTTTACGGCGAAGACGGT	997	53
		PG R	TTTTCTGGTGCATTGGCACG		
*Enolase*	*eno*	E F	CAAGAGCGATCATTGCACGG	201	54
		E R	CATTCGGACGCGGTATGGTA		
*Adhesin Pav*	*pav*	AP F	CCTGTCGGGCGCTTTTATTG	232	56
		AP R	TCCCGGAAGAAGAGTACGGT		
*Adhesin PsaA*	*PsaA*	APSA F	GTTGCAACAGCTGGACACAG	180	54
		APSA R	ATACGGTTGAGTTGGGCTGG		
*Adhesin cluster 1*	*adhC1*	adhC1 F	TTGGGCACATCAGACTGGAC	264	54
		adhC1 R	AGCATCATCAGCTGCCAAGT		
*Adhesin cluster 2*	*adhC2*	adhC2F	CTGCGAGTGGCATCTCCATT	160	52
		adhC2 R	TCAACACTGCGACCTTCTGT		
*Adhesin*	*AF*	AF F	CAGCCAGCACCAGGTTATGA	358	54
		AF R	CTCCTGCGTTGACATGGACT		
*capsule gene* *cluster A*	*CGC A*	1020-F	ACCTTCACTTGCATTCATAGGGT	304	54
		1323-R	TTGTCCCAGAGGGTTCTCCT		
*capsule gene* *cluster B*	*CGC B*	851-F	TAGGAGGTGTTCCTGGGAGG	549	54
		1399-R	TGTCCCACTCCTACTGTCGT		
*capsule gene* *cluster C*	*CGC C*	6329-F	AAAAACGGAGGGCAACAAGC	785	60
		7175-R	CACTTGTACAGGCCACTGGT		
*capsule gene* *cluster D*	*CGC D*	5358-F	TGGAGGGTATTGCCTACCGA	650	54
		6007-R	CCACAGCAGCTTCTTCACCT		
*conserved* *hypothetical protein*	*CHP*	CHP F	CTGCTGATCAAGTCCAAGC	303	52
		CHP R	GAGAAACGACCTTAGCTCCA		
*exopolysaccharide A*	*EpsA*	EpsA F	TTATAGCCTCCCCAGTTTACAC	299	52
		EpsA R	TTTAGCAGTCTCGTCTGCAATC		
*exopolysaccharideB*	*EpsB*	EpsB F	CGCAAGTGCTAATCTAGCTG	317	52
		EpsB R	AGAGAGGCGGAGTATCAATC		
*exopolysaccharide C*	*EpsC*	EpsC F	TAACAACTATCACTGCGACTCC	343	52
		EpsC R	TCAGGGTTCTCAATGATTCCAC		
*exopolysaccharide D*	*EpsD*	EpsD F	TTTCTTATTGCGGCTGCATTGC	270	52
		EpsD R	CTCATCAATTGAGTGTCGTCTG		
*exopolysaccharide L*	*EpsL*	EpsL F	ACCAATCGTACAGATCAACG	473	52
		EpsL R	CTTGAGCCACCACTATCAAG		
*exopolysaccharide R*	*EpsR*	EpsR F	TTTTACCACCGGCTAAAGGAAC	211	52
		EpsR R	TTGCAGAACTGTCATTAGGCTC		
*exopolysaccharide X*	*EpsX*	EpsX F	TATTGAAGCAACAGCCTCACTG	198	52
		EpsX R	TTTTTGTCTGGGTAACTAGCCC		
*rhamnosyltransferase*	*RIF*	RIF F	TTGATGGTAAATCCTGATGG	307	52
		RIF R	GAACAAACCGACCTACAACA		
*30SrRNA gene*	*30S*	30S F	TACGAACACCGTATCCTTGAC	207	52
		30S R	TACGAACACCGTATCCTTGAC		
*LPxTG-1*	*LP1*	LP1-F	GTGAACGTGGAGCTTCCAGA	878	54
		LP1-R	CCACTCACATGGGGGAGTTC		
*LPxTG-2*	*LP2*	LP2-F	GCCAGTGAGAGAACCGTTGA	767	54
		LP2-R	CAGGTTCAAGTGCAACTGCC		
*LPxTG-3*	*LP3*	LP3-F	TTAAGCACAACGGCAACAGC	231	54
		LP3-R	CACGCGAAATGATGGTGCAT		
*LPxTG-4*	*LP4*	LP4-F	GGGAGCACCGGATTCACTTT	928	52
		LP4-R	ACAAAGCCGCAGACCTTACA		

**Table 2 microorganisms-13-01674-t002:** Biochemical results for 9 *L. lactis* isolates.

Number	Ribose	Sucrose	Lactose	Liquid Gelatin	Sorbitol	Maltose	Esculin	VP	Galactose	Trehalose	Glucose
*LL1*	-	+	+	-	-	+	+	+	+	+	+
*LL2*	-	+	+	-	-	+	-	+	+	+	+
*LL3*	-	-	+	-	-	+	+	+	+	+	+
*LL4*	-	+	+	-	-	+	+	+	+	+	+
*LL5*	-	+	+	-	-	+	+	+	+	+	+
*LL6*	-	+	+	-	-	+	+	+	+	+	+
*LL7*	-	-	+	-	-	+	+	+	+	+	+
*LL8*	-	-	+	-	-	+	+	+	+	+	+
*LL9*	-	-	+	-	-	+	+	+	+	+	+

**Table 3 microorganisms-13-01674-t003:** Number of isolates at each MIC value for antimicrobial agents against *L. lactis*.

MIC (μg/mL)														
Antimicrobial	>16	16	8	4	2	1	0.5	0.25	0.12	0.06	0.03	MIC50 (μg/mL)	MIC90 (μg/mL)	Resistance Rate
Penicillin	0	0	0	0	0	1	3	4	0	0	0	0.25	0.50	100
Cephalexin	1	6	1	0	0	0	0	0	0	0	0	16	16	100
Ampicillin	0	0	0	0	0	1	2	3	2	0	0	0.25	0.50	37.5
Ceftiofur	0	1	0	0	0	3	4	0	0	0	0	0.5	1.0	12.5
Cefquinome	0	0	0	0	1	0	1	3	3	0	0	0.25	0.50	0
Lincomycin	1	0	1	0	0	3	1	1	0	1	0	1	1	50
Oxytetracycline	1	2	3	2	0	0	0	0	0	0	0	8	16	100
Marbofloxacin	0	0	0	0	0	6	2	0	0	0	0	1	1	0
Rifaximin	8	0	0	0	0	0	0	0	0	0	0	>16	>16	100
Vancomycin	0	0	0	0	0	0	0	7	1	0	0	0.25	0.25	0

## Data Availability

The original contributions presented in this study are included in the article. Further inquiries can be directed to the corresponding author.

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
