# Peer review of "Virulence, Antibiotic Resistance and Cytotoxic Effects of Lactococcus lactis Isolated from Chinese Cows with Clinical Mastitis on MAC-T Cells"

_microorganisms, 2025, doi:10.3390/microorganisms13071674_

Round 1
Reviewer 1 Report
Comments and Suggestions for Authors
The comments are attached

Author Response
|
3. Point-by-point response to Comments and Suggestions for Authors |
|
Comments 1: [ The abstract is well written but lacks recommendations.] |
|
Response 1: [The recommendations were supplemented on the line 26-28. ] Thank you for pointing this out. We agree with this comment. |
|
Comments 2: [Introduction needs proofreading for typographical errors, including issues with commas, spacing, periods, Italicizing Latin names, and other punctuation that may be incorrectly placed.] |
|
Response 2: Thank you for pointing this out. We have accordingly modified typographical errors in the full text. |
|
Comments 3: [From how many farms and from how many dairy cows? You have mentioned number provinces and number milk samples collected. But not number of farms and number of animals from which milk samples are collected. How about the breeds of dairy cows?] |
|
Response 3: [The samples were collected from 5 farms and from 407 Holstein dairy cows.] |
|
Comments 4: [From each farm how many milk samples were collected? |
|
Response 4: [40 samples were collected from Anhui farm. 56 samples were collected from Gansu farm. 11 samples were collected from Guizhou farm. 262 samples were collected from Ningxia farm. 38 samples were collected from Shandong farm.] |
|
Comments 5: [The sample collection procedure is not easily understandable and needs clarification. Is it composite or quarter samples?] |
|
Response 5: [CM samples were aseptically collected from individual quarters by the authors of this study or trained on-farm personnel. The CM samples for each farm were collected within a 7 day time span. The samples were quickly frozen (-20 |
|
Comments 6: [Did you test the milk with California Mastitis Test before sampling?] |
|
Response 6: We didn’t test the milk with California Mastitis Test before sampling because all the samples were clinical mastitis samples. |
|
Comments 7: [Line 103-104, ‘’Specifically, 10 μl was inoculated onto 5% defibrinated sheep blood agar plates and incubated at 37 °C for 24 hours.’’ What is you base to incubate at 37 °C for hours? The optimal growth of this bacteria is between 25-30 °C.] |
|
Response 7: We chose 37 °C because the normal body temperature of dairy cows is usually between 38.5℃ and 39.5℃, and 37 °C is close to it. |
|
Comments 8: [the text needs proofreading for typographical errors, including issues with commas, spacing, periods, Italicizing Latin names, and other punctuation that may be incorrectly placed.] |
|
Response 8: Thank you for pointing this out. We have accordingly modified typographical errors in the full text by using English language editing services of MDPI. |
|
Comments 9: [What is the prevalence rate of L. lactis in this study?] |
|
Response 9: The prevalence rate of L. lactis is 1.9% in this study. This date resulting from all the milk samples were clinical mastitis samples. We have reasons to doubt that prevalence rate of L. lactis in subclinical mastitis might be even higher. |
|
Comments 10: [Please revise the conclusion to reflect the main findings of your study. Additionally, it is important to outline specific recommendations for stakeholders based on the current results, as well as to propose suggestions for future research endeavors.] |
|
Response 10: The conclusion has been revised on line 506-517.. |
|
|
|
4. Response to Comments on the Quality of English Language |
|
Response : We have accordingly modified typographical errors in the full text by using English language editing services of MDPI. |
Reviewer 2 Report
Comments and Suggestions for Authors
Introduction:
The topic addressed is both timely and relevant, namely the potential pathogenic role of Lactococcus lactis in cases of clinical bovine mastitis. However, the introduction displays several contextual and descriptive weaknesses that limit its communicative effectiveness. Firstly, although antimicrobial resistance is mentioned, it is addressed in a superficial and poorly structured manner. No information is provided regarding the antibiotic classes commonly used in mastitis treatment, nor are the most prevalent genetic mechanisms of resistance in pathogenic bacteria described. Furthermore, there is no quantitative overview of the antimicrobial resistance phenomenon in the livestock sector, and no specific data are cited relating to the Chinese or international context.
Secondly, the article fails to present a comprehensive overview of the most common aetiological agents of bovine mastitis, focusing almost exclusively on Lactococcus spp.. A general description of major pathogens such as Staphylococcus aureus, Streptococcus agalactiae, Streptococcus uberis, Escherichia coli, and Klebsiella spp. is lacking, and no distinction is made between environmental and contagious pathogens, which would be helpful to understand the infection dynamics on farms.
Another critical issue is the absence of references to alternative therapeutic approaches to antibiotics, such as the use of natural antimicrobial compounds. In this regard, recent studies have highlighted the effectiveness of essential oil from Thymus vulgaris (TEO) in combating mastitis-associated bacteria, owing to its capacity to inhibit biofilm formation and reduce bacterial viability. Of particular relevance is the study by Galgano et al. (2025), which investigated the in vitro activity of TEO against pathogens isolated from ruminants affected by mastitis (Pathogens, 14(5), 412. https://doi.org/10.3390/pathogens14050412).
Line 37: There is a redundancy due to repetition of the word “including”: “streptococci-like bacteria including, including Streptococcus, Enterococcus, Aerococcus”.
Line 40: The phrase “as being associated” is redundant and awkwardly constructed. Furthermore, scientific names of bacteria should be italicised.
Line 61: In “genes (epsR, X, A, B, C, D, L)...”, gene names should be italicised.
Materials and Methods:
There is no description of the other bacterial isolates obtained from the 457 clinical mastitis cases. This omission hinders the evaluation of the clinical and epidemiological relevance of L. lactis in bovine mastitis. I ask the authors whether it would be possible to report the other strains isolated.
Analysis of pathogenicity genes is incomplete: the gene list is disorganised and contains repetitions (e.g., NADHO appears twice). No justification is provided for the selection of these 30 genes. It is unclear whether they represent all known virulence genes or just a subset.
Reference to lines 91–94: “This study investigated L. lactis strains obtained from commercial dairy farms across five provinces in China... A total of 457 clinical mastitis (CM) milk samples were collected...” – this sentence is overly long and difficult to read; I recommend splitting it into two separate sentences for clarity.
Repetition and disorganisation in the gene list (lines 172–173): “NADHO, NADHO, SOD Phosphoglucomutase (pgm), Pav, PsaA, eno... LP1, LP2, LP3, LP4” – NADHO is listed twice; the list appears poorly curated.
Line 141: “UV spectrophotometer ((Shimadzu Corporation” contains a double parenthesis.
Line 242: “2 mL of 1 (v/v) Triton” should be corrected to “2 mL of 1% (v/v) Triton X-100”.
Line 254: “37S℃” should be corrected to “37 °C”.
Results:
In section 3.2, the amplification of the 16S rRNA gene and the subsequent phylogenetic analysis are described. However, in Figure 2A, only six samples are shown, whereas the total number of Lactococcus lactis strains tested in the study is nine, as stated in multiple sections of the manuscript. Could the authors clarify why only six strains are shown in Figure 2A and specify whether the remaining three were not amplified, not visualised in the gel, or excluded for other reasons?
Additionally, there is no positive control (a known L. lactis strain) included in the gel lanes, which would be helpful to confirm the expected band position (~1500 bp) and to facilitate visual validation of the results.
Was a positive reference control included in the 16S PCR? If not, could the authors justify this methodological choice and indicate any precautions taken to prevent false positives or ambiguities in band identification?
In section 3.6, the presence of virulence genes in Lactococcus lactis strains was analysed via multiplex PCR, with results shown in Figure 4. However, this section also lacks critical details for result validation.
Positive controls: Could the authors clarify whether positive controls were used for each target gene? Including such controls is essential to validate the efficiency and specificity of PCR amplifications, particularly when dealing with a large set of 30 virulence genes, which may involve highly conserved sequences or be subject to cross-reactivity.
Amplification specificity: Given the multiplex nature of the PCR and the high number of targets, please provide further details on how non-specific amplifications or false positives were ruled out.
In section 3.5, concerning antimicrobial susceptibility testing, the authors mention the use of the Staphylococcus aureus ATCC 29213 quality control strain and refer to CLSI M100-M07 (2019) guidelines. However, it is not clearly stated which criteria or breakpoints were adopted to interpret MIC values for the nine Lactococcus lactis strains.
Since L. lactis is not a standardised microorganism in CLSI or EUCAST guidelines regarding antimicrobial breakpoints, it would be helpful to clarify:
Which guidelines or standards were used to classify isolates as susceptible or resistant;
Whether EUCAST or other recognised sources were also consulted;
How the clinical interpretation of MIC data was managed.
Discussion and Conclusions:
In the discussion section, it would be appropriate to include a specific subsection addressing mastitis prevention and control measures. Referring to international guidelines, such as European recommendations, would provide a more comprehensive and field-applicable perspective.
In particolare, si raccomanda di sottolineare l'importanza delle misure di biosicurezza, dell'igiene e della disinfezione delle strutture e dell'isolamento tempestivo degli animali infetti per limitare la diffusione del patogeno. Inoltre, sarebbe utile menzionare approcci alternativi all'uso degli antibiotici, come l'applicazione di sostanze naturali ad attività antimicrobica, che possono contribuire a contrastare la resistenza e ridurre i tempi di sospensione.
Infine, è importante sottolineare la necessità di preservare l’efficacia degli antibiotici attraverso un uso prudente e mirato, per la tutela della salute degli animali, della sicurezza alimentare e della salute pubblica.

Author Response
|
3. Point-by-point response to Comments and Suggestions for Authors |
|
Comments 1: [Firstly, although antimicrobial resistance is mentioned, it is addressed in a superficial and poorly structured manner. No information is provided regarding the antibiotic classes commonly used in mastitis treatment, nor are the most prevalent genetic mechanisms of resistance in pathogenic bacteria described. Furthermore, there is no quantitative overview of the antimicrobial resistance phenomenon in the livestock sector, and no specific data are cited relating to the Chinese or international context.] |
|
Response 1: [The information of the antibiotic classes commonly used in mastitis treatment and mechanisms of resistance in pathogenic bacteria described was supplemented on the line 75-82. The content of genetic mechanisms of resistance was supplemented on line 82-87.The overview of the antimicrobial resistance phenomenon Chinese was supplemented on line 85-89.] Thank you for pointing this out. We agree with this comment. |
|
Comments 2: [Secondly, the article fails to present a comprehensive overview of the most common aetiological agents of bovine mastitis.] |
|
Response 2: [The overview of the most common aetiological agents of bovine mastitis was supplemented on the line 34-37.] Thank you for pointing this out. We agree with this comment. Limited by the word count of the MDPI magazine, we are unable to elaborate on more aspects. |
|
Comments 3: [Another critical issue is the absence of references to alternative therapeutic approaches to antibiotics, such as the use of natural antimicrobial compounds.] |
|
Response 3: [References to alternative therapeutic approaches to antibiotics, including the use of natural antimicrobial compounds was supplemented on the line 91-95.] Thank you for pointing this out. We agree with this comment. Limited by the word count of the MDPI magazine, we are unable to elaborate on more aspects. |
|
Comments 4: [There is a redundancy due to repetition of the word “including”: “streptococci-like bacteria including, including Streptococcus, Enterococcus, Aerococcus”.] |
|
Response 4: [The repetition of the word “including” has been revised] Thank you for pointing this out. We agree with this comment. |
|
Comments 5: [The phrase “as being associated” is redundant and awkwardly constructed. Furthermore, scientific names of bacteria should be italicised.] |
|
Response 5: [The phrase “as being associated” has been revised to “associated” on line 44. Scientific names of bacteria have been revised to italicized in the full text.] Thank you for pointing this out. We agree with this comment. |
|
Comments 6: [Line 61: In “genes (epsR, X, A, B, C, D, L)...”, gene names should be italicised] |
|
Response 6: [Gene names have been revised to “italicised” on line 65 and the full text.] Thank you for pointing this out. We agree with this comment. |
|
Comments 7: [There is no description of the other bacterial isolates obtained from the 457 clinical mastitis cases.] |
|
Response 7: [We isolated other bacteria but we did not descript that in the text. The coagulase negative staphylococci, Streptococcus agalactiae, Streptococcus dysgalactiae, Staphylococcus aureus, Streptococcus Paris, Enterococcus faecalis, Lactococcus garvieae, Aerococcus viridans, Bacillus,Arcanobacterium and some Gram-negative bacterium were isolated. These bacteria are considered to be the main causes of mastitis in dairy cows compared with Lactococcus lactis, and many studies have been conducted in depth and detail. The aim of this study was to focus on L. lactis. Everyone of these nine L. lactis strains are the single pathogenic microorganisms in samples and cause clinical mastitis alone.] Thank you for pointing this out. We agree with this comment but limited by the word count of the MDPI magazine, we are unable to elaborate on more aspects. |
|
Comments 8: [Analysis of pathogenicity genes is incomplete: the gene list is disorganised and contains repetitions (e.g., NADHO appears twice). No justification is provided for the selection of these 30 genes. It is unclear whether they represent all known virulence genes or just a subset.] |
|
Response 8: [Thank you for pointing this out. The repetition of “NADHO” has revised on line 187. It is true that Analysis of pathogenicity genes is incomplete, however, these genes were classified into some main types of virulence, including Hemolysin, Adhesin, NADH, Superoxide ,capsule and so on. It is designed to ennable people to have a preliminary understanding of the virulence of this new pathogen. In the subsequent research, we plan to conduct a detailed and segmented study. Follow your instructions we revised the Classification sequence in Table 1 .] |
|
Comments 9: [Reference to lines 91–94: “This study investigated L. lactis strains obtained from commercial dairy farms across five provinces in China... A total of 457 clinical mastitis (CM) milk samples were collected...” – this sentence is overly long and difficult to read; I recommend splitting it into two separate sentences for clarity.] |
|
Response 9: [This sentence have been revised on line 98-99 ] Thank you for pointing this out. We agree with this comment. |
|
Comments 10: [Line 141: “UV spectrophotometer ((Shimadzu Corporation” contains a double parenthesis.] |
|
Response 10: [The double parenthesis in this sentence have been revised on line 142-143 ] Thank you for pointing this out. |
|
Comments 11: [Line 242: “2 mL of 1 (v/v) Triton” should be corrected to “2 mL of 1% (v/v) Triton X-100”.] |
|
Response 11: [This sentence have been revised on line 238] Thank you for pointing this out. |
|
Comments 12: [Line 254: “37S℃” should be corrected to “37 °C”.] |
|
Response 12: [This sentence have been revised on line 249] Thank you for pointing this out. |
|
Comments 13: [In section 3.2, the amplification of the 16S rRNA gene and the subsequent phylogenetic analysis are described. However, in Figure 2A, only six samples are shown, whereas the total number of Lactococcus lactis strains tested in the study is nine, as stated in multiple sections of the manuscript. Could the authors clarify why only six strains are shown in Figure 2A and specify whether the remaining three were not amplified, not visualised in the gel, or excluded for other reasons?] |
|
Response 13: [We tested nine Lactococcus lactis strains in the study. A part of this experiment was practiced at the early stage when the samples were transported and received at different times due to COVID-19 restrictions so we did not test all the samples at once. The research focused on virulence, antibiotic resistance and cytotoxic effects of Lactococcus lactis. In the 16S rRNA identification and homology analysis detection, the amplification of a band approximately 1500 bp in length was successful. The 9 results were submitted to NCBI for comparison, which identified the organism as L. lac-tis. The Lactococcus lactis strains were well preserved and we can also take time to redo this test if you insist.] Thank you for pointing this out. |
|
Comments 14: [Additionally, there is no positive control (a known L. lactis strain) included in the gel lanes, which would be helpful to confirm the expected band position (~1500 bp) and to facilitate visual validation of the results. Was a positive reference control included in the 16S PCR? If not, could the authors justify this methodological choice and indicate any precautions taken to prevent false positives or ambiguities in band identification?] |
|
Response 14: [The amplified product showing a band at 1500 bp was sent to Sangon Biotech (Shanghai) for sequencing. The assembled sequence was then uploaded to the National Center for Biotechnology Information (NCBI) website for BLAST analysis. Strains exhibiting greater than 99% similarity in their 16S rRNA gene sequence were classified as the same strain. Nine sequence results demonstrated a match to Lactococcus lactis in GenBank.] Thank you for pointing this out. |
|
Comments 15: [In section 3.6, the presence of virulence genes in Lactococcus lactis strains was analysed via multiplex PCR, with results shown in Figure 4. However, this section also lacks critical details for result validation. Positive controls: Could the authors clarify whether positive controls were used for each target gene? Including such controls is essential to validate the efficiency and specificity of PCR amplifications, particularly when dealing with a large set of 30 virulence genes, which may involve highly conserved sequences or be subject to cross-reactivity. Amplification specificity: Given the multiplex nature of the PCR and the high number of targets, please provide further details on how non-specific amplifications or false positives were ruled out.] |
|
Response 15: [We sent the amplification products of each band, whose base pair count matched that of the corresponding virulence gene, to Shanghai Sangon Biotech Co., Ltd. for sequencing. The assembled sequencing results were then uploaded to the NCBI (National Center for Biotechnology Information) website for BLAST alignment. Samples with results that matched the corresponding gene sequences were determined to be positive. Therefore, the positive samples within these bands were all confirmed as genuine positives.The rationale for not selecting alternative positive controls is twofold. Primarily, our objective was to conduct preliminary screening of virulence genes in Lactococcus lactis for pathogenicity validation. Secondly, the PCR experiment was designed to identify the most virulent strain among our isolated pathogenic bacteria for subsequent research. Consequently, we exclusively performed experiments on the nine-strain bacterial sample. These Lactococcus lactis have some pecial features and was considered as emerging pathogen of Bovin Clinical Mastitis. So we chose the 30 virulence genes which play important roles in mastitis Refer to other major pathogenic bacteria] Thank you for pointing this out. |
|
Comments 16: [Which guidelines or standards were used to classify isolates as susceptible or resistant; Whether EUCAST or other recognised sources were also consulted; How the clinical interpretation of MIC data was managed.] |
|
Response 16: [We first adopt the CLSI standard. If it is not available, we will refer to EUCAST. The antibiotics in this study are commonly used in dairy farms, as well as vancomycin and other antibiotics that are very important to humans. If the common antibiotics resistance occurs to Lactococcus lactis, it is advisable to consider switching to other sensitive antibiotics for treatment.] Thank you for pointing this out. |
|
Comments 17: [In the discussion section, it would be appropriate to include a specific subsection addressing mastitis prevention and control measures. Referring to international guidelines, such as European recommendations, would provide a more comprehensive and field-applicable perspective. In particular, it is recommended to include the importance of biosecurity measures, hygiene and disinfection of facilities, and timely isolation of infected animals to limit the spread of the pathogen. Furthermore, it would be valuable to mention alternative approaches to antibiotic use, such as the application of natural substances with antimicrobial activity, which may help counteract resistance and reduce withdrawal times. Finally, it is important to stress the need to preserve antibiotic efficacy through prudent and targeted use, for the protection of animal health, food safety, and public health.] |
|
Response 17: [A specific subsection addressing mastitis prevention and control measures have been supplemented on line 493-518.] Thank you for pointing this out. |

Reviewer 3 Report
Comments and Suggestions for Authors
General comments:
This study evaluates and characterize the presence of Lactobacillus lactis in mastitic milk of dairy cows, his antimicrobial resistance, virulence gene expression, and their adhesion and invasion in cells. The study is novel, and well done overall without major concerns. Nonetheless some improvement is required as a final version. As we know, Lactobacillus lactis is considered a normal microbiota of the mammary gland, and it has been suggested as a relevant bacteria modulator (even for treating mastitis using some strains) (e.g., see https://doi.org/10.1186/s13567-017-0429-2). I suggest to discuss this duality (microbiota vs. pathogens). A mention can also be done in the introduction. The part of indiscriminate use of antimicrobial should be revised according the recent norms. Please add the objectives to the introduction and abstract. Some additional references in the methodologies can clearly ensure the replicability of the analysis. The results are clearly presented and supported by useful figures and tables. A sightly improvement of the discussion can be done regarding the previous comment on the duality of the L. lactis.
Specific comments:
L41: spp.; please italicize genus and species. Please use space appropriately. Check the whole manuscript.
L71: I_ suggest to add “…has become, for long time,…”. Today several restrictions of antimicrobials were made in some world regions.
L81-89: this is a discussion. Please clearly add the objectives of the study and move this part to the discussion section.
L90: some references for the methodologies reported in M&M are welcome to improve the reproducibility of the study.
L92. How may dairy farms? What are the inclusion criteria for sample selections.
L134: you don’t need to include “(S. aureus)”.
L314: “Table 2”.
L327: Please add “hours or h.” to the x-axis.
L499: This value was not reported before.
Author Response
|
3. Point-by-point response to Comments and Suggestions for Authors |
|
Comments 1: [I suggest to discuss this duality (microbiota vs. pathogens). The part of indiscriminate use of antimicrobial should be revised according the recent norms. Please add the objectives to the introduction and abstract.] |
|
Response 1: [This part of duality and references were supplemented on line 421-428. The part of indiscriminate use of antimicrobial has be revised on the line 75-80, and the line 82-95. The objectives were add in the introduction and abstract on line 24-28.] Thank you for pointing this out. We agree with this comment. |
|
Comments 2: [L41: spp.; please italicize genus and species. Please use space appropriately. Check the whole manuscript.] |
|
Response 2: [The genus and species were italicized in the whole manuscript.] Thank you for pointing this out. We agree with this comment. |
|
Comments 3: [L71:I suggest to add “…has become, for long time,…”. Today several restrictions of antimicrobials were made in some world regions.] |
|
Response 3: [This less rigorous sentence has been delete.] Thank you for pointing this out. We agree with this comment. |
|
Comments 4: [L81-89: this is a discussion. Please clearly add the objectives of the study and move this part to the discussion section] |
|
Response 4: [The objectives of the study has been add, see line 93-95. The revised content has been moved to the discussion section, see line 493-505.] Thank you for pointing this out. We agree with this comment. |
|
Comments 5: [L90: some references for the methodologies reported in M&M are welcome to improve the reproducibility of the study] |
|
Response 5: [The relevant references has been cited, see line 99,119,143 and so on.] Thank you for pointing this out. We agree with this comment. |
|
Comments 6: [L92: How may dairy farms? What are the inclusion criteria for sample selections.] |
|
Response 6: [There 5 farms the clinical mastitis sample was collected. Abnormal milk (e.g., clots, flake, and watery milk) was identified by farm personnel as CM samples. CM samples were aseptically collected from individual quarters by the authors of this study or trained on-farm personnel.] |
|
Comments 7: [L134: you don’t need to include “(S. aureus)”.] |
|
Response 7: [The “(S. aureus)”was chosen just as an control group.] |
|
Comments 8: [L314: “Table 2”.] |
|
Response 8: [This mistake has been corrected on line 306] Thank you for pointing this out. |
|
Comments 9: [L327: Please add “hours or h.” to the x-axis.] |
|
Response 9: [This typographical errors has been corrected on line 318] Thank you for pointing this out. |
|
Comments 10 [L499: This value was not reported before..] |
|
Response 10: [This description has been removed.] Thank you for pointing this out. |
|
|
|
4. Response to Comments on the Quality of English Language |
|
Response :The English corrections were done with English language editing services of MDPI. |
